# Analysis of Carbonation Behavior of Cracked Concrete

**DOI:** 10.3390/ma15134518

**Published:** 2022-06-27

**Authors:** Qun Guo, Lexin Jiang, Jianmin Wang, Junzhe Liu

**Affiliations:** 1College of Architecture Engineering, Qingdao Agricultural University, Qingdao 266109, China; 20202204007@stu.qau.edu.cn; 2College of Civil and Environmental Engineering, Ningbo University, Ningbo 315211, China; jianglexin@nbu.edu.cn

**Keywords:** carbonation, cracks, durability, cement paste

## Abstract

The crack and carbonation of concrete pose a great challenge to its durability. Therefore, this paper studies the effect of cracks on the carbonation depth of cement paste under different factors. The relationship between carbonation and cracks was determined, and the carbonation mechanism of cement paste with cracks was clarified. The results show that a small water–binder ratio can effectively inhibit the carbonation process. The bidirectional carbonation enlarged the carbonation area around the crack. Within 21 days of the carbonation, the carbonation depth increased with carbonation time, and the Ca(OH)_2_ on the surface of the specimen was sufficient, allowing for a convenient chemical reaction with CO_2_. The influence of crack width on the carbonation process at the crack was greater than the influence of the crack depth. Carbonation influenced the hydration of cement-based materials, altering the types and quantities of hydration products. In conclusion, accurately predicting the regularity of carbonation in cracked structures is critical for improving the durability of concrete.

## 1. Introduction

Most concrete structures have cracks caused by shrinkage, the greenhouse effect, or load [1,2]. These cracks become channels for harmful substances to invade the interior of the concrete. It accelerates the failure process of the protective layer of concrete and causes steel corrosion, thereby reducing the durability of the concrete structure [3,4,5]. Carbonation is one of the critical factors causing steel corrosion [6,7,8]. The diffusion of CO_2_ into concrete for carbonation can be divided into four steps, as shown in Figure 1. The existence of cracks promotes the diffusion of CO_2_ in concrete [9].

The research on concrete cracking is relatively extensive at home and abroad. It mainly focuses on the causes of cracks, crack control and numerical simulation, characteristics of cracks, and so on [10,11,12,13]. However, there are relatively few experimental studies on the carbonation law of concrete at the crack [14,15,16,17]. Miao et al. [18] found that micro cracks and micropores in concrete were important factors affecting the carbonation performance of concrete. In this respect, Alahmad et al. [19] systematically analyzed the influence of cracks on CO_2_ transport and studied the carbonation process and carbonation model of cracked concrete. Jin et al. [20] showed that the crack had little effect on the carbonation depth around the concrete crack when the width of the shrinkage joint was less than 0.1 mm. Furthermore, with the increase in crack width, the carbonation depth around the crack increased as a quadratic function. Bogas et al. [21] pointed out that according to the relationship between carbonation depth and crack length, the estimated carbonation rate under actual exposure conditions can increase the carbonation rate of cracked concrete by more than 80%. Carbonation has always been one of the world’s difficulties with the durability of concrete, and it is also the most important problem to be solved at present. The existence of cracks in concrete durability research has a very vital significance.

The goal of this research is to decipher the carbonation law of cracked cement paste and to investigate the carbonation mechanism. In order to accomplish the goal, a method of non-destructive preparation of cracks was used to obtain the crack width and depth required for the test. In addition, the cracked cement paste specimens were carbonated. On this basis, the relationship between carbonation and cracks was determined using phenolphthalein and XRD. It provides an effective theoretical foundation for the future study of predicting the life of concrete structures.

## 2. Materials and Methods

### 2.1. Materials and Preparation of Specimens

The test used P • O 42.5 normal Portland cement (produced by Zhejiang Ningbo conch Co., Ltd., Ningbo, China). The basic chemical composition is shown in Table 1. A number of 50 mm × 50 mm × 50 mm cement paste specimens with water–binder ratios of 0.30, 0.35, and 0.40 were made. Three specimens were made for each water–binder ratio to measure the carbonation depth at different ages. Before pouring, 0.2-mm-thick copper was cut into 40 mm × 25 mm sections, and its surface was lubricated with lubricating oil. After the specimen was vibrated and screeded, the copper was inserted into the specimen. Before setting and hardening, the copper was removed to obtain the specimen, as shown in Figure 2. The size of the crack was 40 mm × 0.2 mm × 15 mm. In addition, different widths and depths of the crack were also needed to study the influence of crack size on the carbonation process. Therefore, when preparing specimens, according to the same method above, 40-mm-wide copper was cut according to the crack depth, whether 10 mm, 15 mm, 20 mm, 25 mm, or 30 mm, and the crack width, whether 0.1 mm, 0.2 mm, or 0.3 mm. The cut coppers were then inserted into fresh paste. The copper was taken out before the paste hardened. The specimens with the required crack size were obtained, and one specimen was made for each size. The crack depth of 15 mm and width of 2 mm was controlled when making double crack specimens. Specimens with water–binder ratios of 0.30, 0.35, and 0.40 were used to measure specimens with crack distances of 10 mm, 20 mm, and 30 mm, respectively.

### 2.2. Carbonation Test

After curing for 28d, the rapid carbonation test of the specimens was carried out according to the GB/T 50082-2009 [22]. The specimens were carbonated for 7 d, 14 d, and 21 d, and broken in a direction perpendicular to the crack. The remaining powder on the crack surface was scraped off. A small sprayer was used to spray the phenolphthalein indicator with a mass fraction of 1% on the crack surface. The carbonated area was colorless, and the noncarbonated area was pink, as shown in Figure 3. Each specimen was measured and the carbonation depth at the crack was recorded, along with 5 mm, 10 mm, 15 mm, 20 mm, and 25 mm to its left and right. According to the measured data, the effects of different factors on carbonation depth were drawn.

### 2.3. XRD Test

The specimens were taken out of the carbonation box at the specified age. The samples 10 mm, 15 mm, and 20 mm away from the crack surface were drilled with a drilling machine. The schematic diagram is shown in Figure 4. The samples were selected according to Figure 4. The sample was sieved through a 0.3 mm fine-pore sieve to obtain a powder with a particle size of less than 0.3 mm. After drying in an electric blast-drying oven at 60 °C for 24 h, the powder was placed in a small plastic sample bag and sealed for standby. During the test, glass slide samples were made for an X-ray powder diffraction (XRD) test. The XRD model was D8Advance Davinci, using Cu Kα1 radiation, and the tube voltage was 40 kV, and the tube current was 40 mA. The scanning method was continuous scanning, and the range was 20°~90°, the rate was 8°/min, the step size was 0.02°.

## 3. Results and Discussion

### 3.1. Carbonation Process of the Single Crack Specimen

#### 3.1.1. Carbonation Area of Single Crack Specimen

Figure 5 is the schematic diagram of carbonation depth. In Figure 5, when the water–binder ratio increases to 0.40, the carbonation influence area at the crack continues to expand around it, and the influence range is between −5 mm and 5 mm. At this time, the carbonation depth at the crack is most obviously affected by the carbonation age. The influence area of carbonation is similar to an inverted “V” shape, and a bidirectional carbonation environment is formed near the crack and the surface of the specimen. CO_2_ intruded in both directions at the same time, increasing the carbonation depth in the 5 mm area near the crack. It can be seen that the concrete with low water–binder ratio can effectively inhibit the carbonation process.

Table 2 shows the magnification factor of the carbonation depth at the crack and the opposite side of the crack under the conditions of different water–binder ratios and carbonation time. Combined with Figure 5 and Table 2, it can be seen that the maximum carbonation depth occurs at the crack. The carbonation depth at the crack is 1.38–3.87 times that of the carbonation on opposite side of the crack. It shows that the existence of cracks promotes the carbonation process.

#### 3.1.2. Effect of Water–Binder Ratio on Carbonation Process of the Single Crack Specimen

Figure 6 shows the relationship between the water–binder ratio and carbonation depth when the crack depth is 15 mm and the width is 0.2 mm. It can be seen from Figure 6 that when the carbonation age is certain, the carbonation depth increases with the increase in the water–binder ratio. The carbonation resistance with a low water–binder ratio is obviously stronger than that with high water–binder ratio, and the carbonation depths at the cracks all exceed the crack depth. Under the condition of a low water–binder ratio, part of CO_2_ can directly enter the cement paste along the crack channel to complete carbonation [23]. Moreover, with carbonation time, the greater the water–binder ratio is, the greater the carbonation depth is. On the one hand, the change in the water–binder ratio affects the hydration of cement. Hydration effects the compactness and porosity of concrete structures. When the water–binder ratio increases, the porosity increases, and the compactness decreases. It is easy for the water and CO_2_ to enter the cement paste because this accelerates carbonation. On the other hand, bidirectional carbonation is formed on the surface of the cement paste. This phenomenon accelerates the carbonation speed. After carbonation for 21 days, the carbonation depth at the crack of the 0.4 water–binder ratio specimen was 1.5 times that of the 0.3 water–binder ratio specimen and 1.38 times that of the 0.35 water–binder ratio.

#### 3.1.3. Effect of Carbonation Time on Carbonation Process of the Single Crack Specimen

Figure 7 shows the relationship between carbonation time and depth when the crack depth is 15 mm and the width is 0.2 mm. When the water–binder ratio is certain, with the increase in carbonation time, the carbonation rate increases and the carbonation depth expands. The Ca(OH)_2_ and CO_2_ on the surface of the specimen are sufficient. Ca(OH)_2_ in cement paste can fully react with CO_2_ to form CaCO_3_. The carbonation depth at the crack is significantly greater than that at the opposite side of the crack. The maximum carbonation depth is visible in the crack’s center. Shun Bo et al. [24] found that the carbonation rate of concrete was faster before 28 days, which is consistent with the results of this study. When the water–binder ratio is 0.4, compared with the other two groups of specimens, the carbonation depth at the crack increases significantly with carbonation time. The carbonation process is affected by Ca(OH)_2_ and compactness. This is due to the complete reaction of CO_2_ in the pores with Ca(OH)_2_ and other substances.

A large number of experimental data show that the carbonation depth of the cement paste specimen with a crack is directly proportional to the cube root of carbonation time, $Y=A\sqrt[{3}]{T}$, where *Y* is the carbonation depth, *T* is the carbonation time, and *A* is the proportional coefficient of $\sqrt[{3}]{T}$. After performing linear regression analysis on the data in Figure 7 by the statistical analysis software SPSS, the coefficient A is obtained, as shown in Table 3 and Table 4. It can be seen from Table 3 and Table 4 that there is a large gap when the water–binder ratio is different. Therefore, the linear relationship equation between the proportion coefficient and the water–binder ratio is established, $A=D\left(W/C\right)+E$, where *W*/*C* is the water–binder ratio, *D* is the proportional coefficient of *W*/*C*, and *E* is the constant. After fitting the coefficient A in Table 3 and Table 4, the proportional coefficient D and constant E are obtained, as shown in Table 5. It is found that the Sig. of each result is less than 0.05, indicating that the fitting results are good. When the crack depth and crack width are known, the carbonation depth of any water–binder ratio specimen in a certain range can be calculated after a certain carbonation time. The relationship is shown in Formulas (1) and (2).

At crack:(1)$$Y=\left(16.850W/C+1.490\right)\sqrt[{3}]{T}$$

At the opposite side of the crack:(2)$$Y=\left(32.100W/C+7.376\right)\sqrt[{3}]{T}$$

#### 3.1.4. Effect of Crack Size on Carbonation Process

Table 6 shows the magnification factor of the carbonation depth at the crack and the opposite side of the crack under the conditions of different crack depths and widths. According to the data in Table 6, when there are cracks on the specimen’s surface, the carbonation depth at the crack is 1.46–2.10 times the carbonation depth on the opposite side of the crack. This is attributed to the fact that cracks can provide a natural channel for CO_2_ to penetrate the specimen to achieve deeper carbonation. It shows that the existence of cracks promotes the carbonation process.

Figure 8 shows the distribution of carbonation depth at cracks with different crack sizes when the water–binder ratio is 0.35 and the carbonation time is 14 d. It can be seen from Figure 8 that when the crack width is certain, the carbonation depth increases with the increase in crack depth. When the crack depth exceeds 25 mm, the carbonation depth gradually tends to be stable, and the threshold appears. The healing function of cracks has played a great role. There are four causes of crack healing, as shown in Figure 9. Among these reasons, the main one is the crack self-healing caused by the further hydration and carbonation of cement. On the one hand, free Ca^2+^ in cement paste reacts with dissolved CO_2_, the generated CaCO_3_ crystals make the pore structure dense. CaCO_3_ reduces the pore structure and even causes crack self-healing [25,26]. The chemical reaction formula is as follows [27]:(3)$$H_{2}O+CO_{2}\Leftrightarrow H_{2}CO_{3}\Leftrightarrow H^{+}+HCO_{3}^{-}\Leftrightarrow 2H^{+}+CO_{3}^{2-}$$
(4)$$Ca^{2+}+CO_{3}^{2-}\Leftrightarrow CaCO_{3}\;(\mathrm{pH}\;\mathrm{water}\;>\;8)$$
(5)$$Ca^{2+}+HCO_{3}^{-}\Leftrightarrow CaCO_{3}+H^{+}\;(7.5\;<\;\mathrm{pH}\;\mathrm{water}\;<\;8)$$ CaCO_3_ also has the ability to reduce alkaline environments [28]. The permeability coefficient of concrete gradually decreases due to crack self-healing [29,30]. It reduces the CO_2_ diffusion and inhibits the process of carbonation. On the other hand, when the crack depth exceeds 25 mm, the CO_2_ in the deep crack cannot be supplemented in time after being exhausted, resulting in the carbonation reaction gradually becoming stable. When the crack depth is certain, with the increase in crack width, the carbonation depth and influence range at the crack continue to expand. This is due to the fact that as the width of the crack increases, the integrity and compactness of the concrete deteriorate, and the formed CaCO_3_ crystals cannot completely fill the cracks and cannot block the transfer of CO_2_ into the concrete. The influence of crack width on the carbonation process at the crack is more important than the crack depth. According to the carbonation depth distribution on the opposite side of the crack in Figure 10, the existence of the crack does not affect the carbonation process on the opposite side of the crack.

### 3.2. Carbonation Process of the Specimen with Double Crack

#### 3.2.1. Carbonation Area of Double Crack Specimen

The shape of carbonation depth of a double-crack specimen is different from that of a single-crack specimen. As shown in Figure 11, when the carbonation time is 14 days, the carbonation depth between the cracks of the double-crack specimen generally presents a “U” shape. When the crack distance is 10 mm, there is a sharp point in the crack center, and the carbonation depth at the sharp point is significantly higher than the carbonation depth on the opposite side of the crack. It shows that the carbonation of specimens between cracks is obviously affected by cracks. When the crack distance is 20 mm, there is a smooth point in the carbonation depth between the cracks. At this time, the effect of cracks on the carbonation depth between cracks is no longer obvious for the specimen with a water–binder ratio of 0.3. However, the carbonation depth at the crack of the specimen with a high water–cement ratio is still higher than the carbonation depth at the opposite side of the crack. When the crack distance is 30 mm, there is a straight section at the “U”-shaped bottom between the cracks. It shows that at this distance, the influence of cracks on the carbonation of the specimen between cracks is no longer obvious. It is concluded that the carbonation between cracks is not obvious when the crack distance is large enough.

#### 3.2.2. Effect of Double Cracks Distance on Carbonation Process

Figure 12 shows that when the water–binder ratio is certain and the carbonation time is 14 days, with the increase in crack distance, the carbonation depth on the opposite side of the crack and the carbonation depth at the crack has no apparent change law within a certain range. It illustrates that the change in the crack distance has no direct impact on the carbonation depth at the crack and the opposite side of the crack, but it has a significant effect on the carbonation depth between cracks. When the water–binder ratio is certain, the carbonation depth between cracks decreases with the increase in the crack distance. When the crack distance is 30 mm, the carbonation depth between cracks tends to be the carbonation depth on the opposite side of the crack. When the crack distance is small, the carbonation depth between the cracks is affected by the carbonation of the left and right cracks and the surface. When the distance between two cracks is relatively close, the carbonation in three directions affects each other, and the carbonation between cracks is more obvious. With the increase in crack distance, the influence of double cracks on carbonation between cracks decreases. When the crack distance is large enough, the carbonation depth of the crack and its surroundings is similar to that of a single crack.

#### 3.2.3. Effect of Water–Binder Ratio on the Carbonation Process of the Double-Crack Specimen

Figure 13 shows the influence of the water–binder ratio on the carbonation depth of double-crack specimens with a carbonation time of 14 days. When the crack distance is certain, the carbonation depth increases with the increase in the water–binder ratio. However, it is worth noting that the carbonation depth of each group does not decrease with the increase in the water–binder ratio; the growth rate of carbonation depth is constantly increasing. Therefore, in practical engineering, choosing low water–binder ratio test parts can effectively inhibit the carbonation as far as possible.

### 3.3. Carbonation Mechanism of Cement Paste Specimen with Cracks

In the specimens with a water–binder ratio of 0.35 and a carbonation time of 7 days, the parts 10 mm, 15 mm, and 20 mm away from the crack surface were ground for XRD test. Figure 14 shows the XRD patterns of the three sample parts. It is observed that there are Ca(OH)_2_ and CaCO_3_ diffraction peaks with different intensities in three places. The Ca(OH)_2_ diffraction peaks disappear gradually with the decrease in crack depth, and CaCO_3_ diffraction peaks increase. Carbonation affects the hydration of cement-based materials and changes the types and amounts of hydration products. The CO_2_ diffused into cement-based materials reacts with Ca (OH)_2_ [31] and responds with hydrated products such as C-S-H, C_3_S, C_2_S, and other gelatinous materials to form CaCO_3_. The distribution of carbonation is clearly seen in Figure 13. When it is 20 mm away from the crack surface, it is difficult to supplement CO_2_ after it is consumed, resulting in insufficient CO_2_ concentration there. The diffraction peak intensity of CaCO_3_ and Ca(OH)_2_ are similar. As the distance from the crack surface is closer, the CO_2_ concentration is sufficient, and CO_2_ forms a bidirectional intrusion pattern along with the crack point. The intensity of the diffraction peaks of Ca(OH)_2_ gradually decreased, and the intensity of the diffraction peaks of CaCO_3_ gradually increased. At this time, phenolphthalein is not colored, indicating that the places 10 mm and 15 mm away from the crack surface are completely carbonated areas.

## 4. Conclusions

(1) The carbonation influence area resembled an inverted “V” shape in single crack specimen and “U” shape in double crack specimen. The formation of a bidirectional carbonation environment increased the carbonation depth.

(2) A small water–binder ratio can effectively inhibit the carbonation process. Within 21 days of the carbonation, the carbonation depth increased with carbonation time.

(3) When the crack depth exceeded 25 mm, the crack self-healed and inhibited the carbonation reaction. However, when the crack width exceeded a certain range, the crack was unable to completely self-heal, and the carbonation depth increased when the crack width increased.

(4) Carbonation influenced the hydration of cement-based materials, altering the types and quantities of hydration products.

## Figures and Tables

**Figure 1 materials-15-04518-f001:**
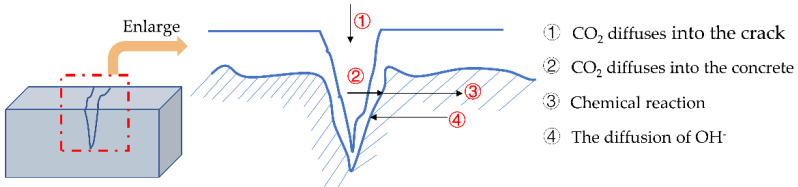
Schematic diagram of concrete carbonation at crack.

**Figure 2 materials-15-04518-f002:**
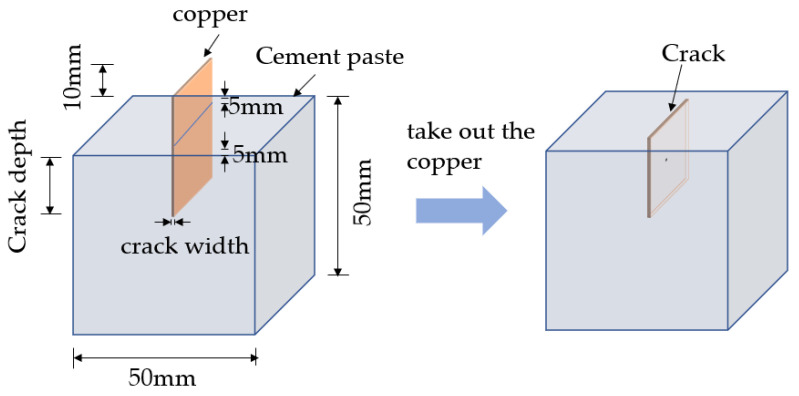
Schematic diagram of cracked specimen.

**Figure 3 materials-15-04518-f003:**
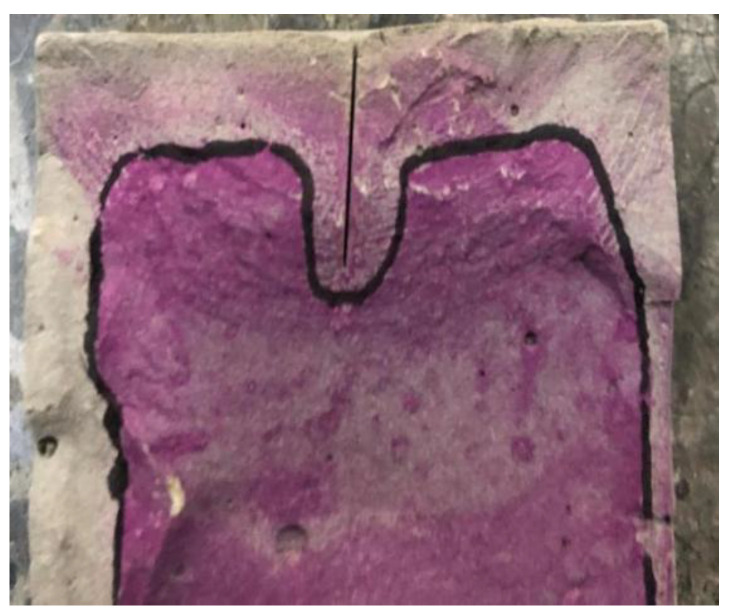
Carbonation distribution diagram.

**Figure 4 materials-15-04518-f004:**
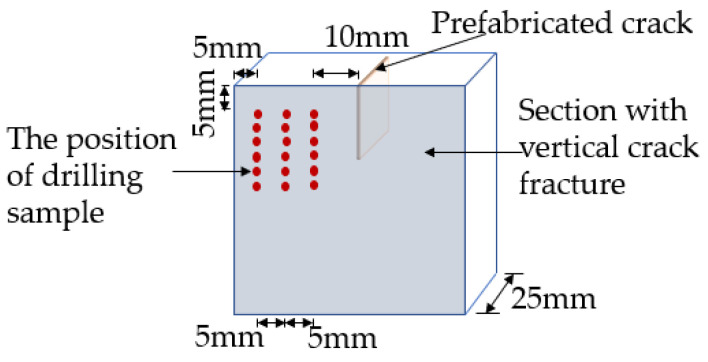
Schematic diagram of drilling for powder.

**Figure 5 materials-15-04518-f005:**
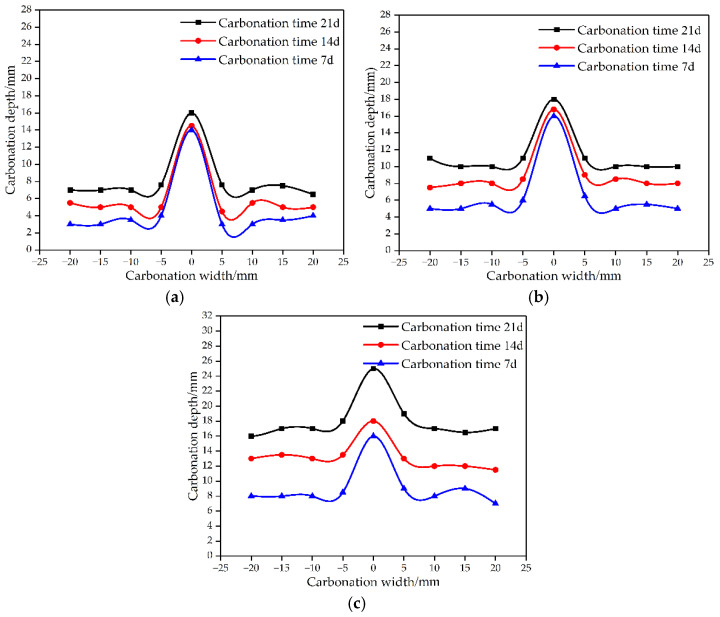
Schematic diagram of carbonation depth. (**a**) W/B 0.30. (**b**) W/B 0.35. (**c**) W/B 0.40.

**Figure 6 materials-15-04518-f006:**
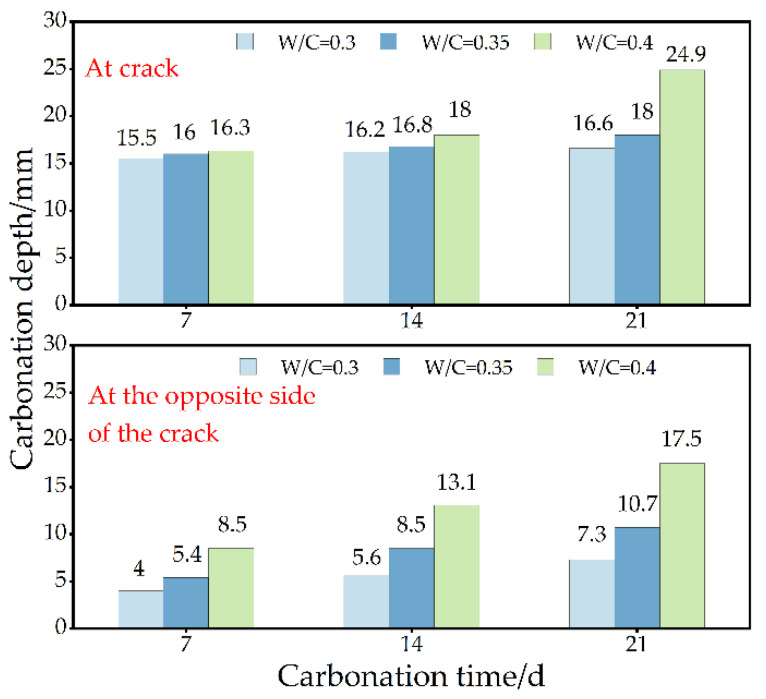
Effect of water–binder ratio on carbonation process.

**Figure 7 materials-15-04518-f007:**
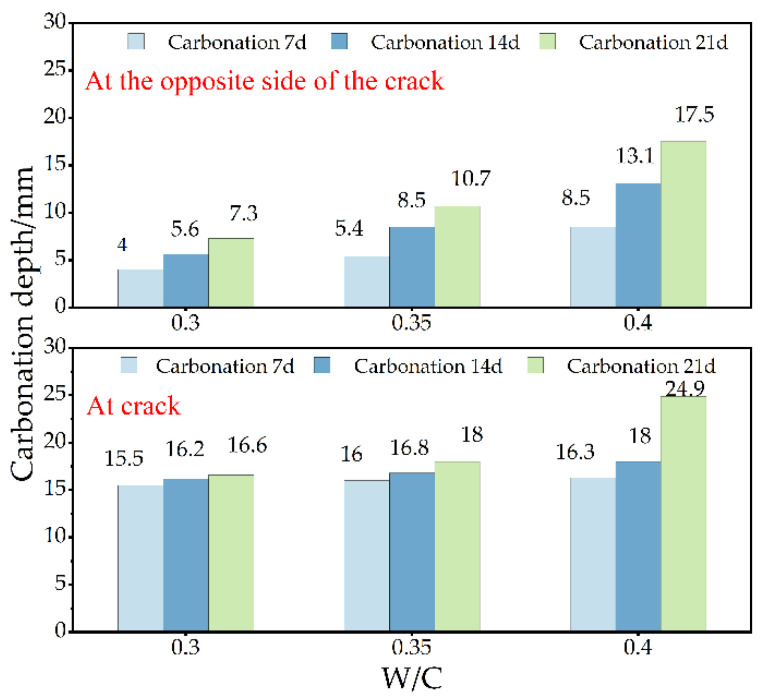
Effect of carbonation time on carbonation process.

**Figure 8 materials-15-04518-f008:**
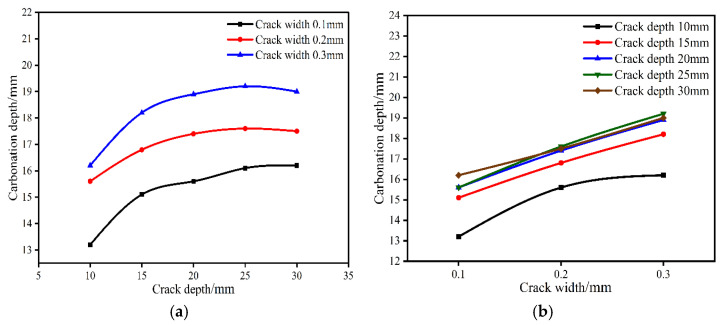
Effect of crack size on carbonation depth at crack when the water–binder ratio is 0.35 and the carbonation time is 14 d. (**a**) Effect of crack depth on carbonation depth at crack. (**b**) Effect of crack width on carbonation depth at crack.

**Figure 9 materials-15-04518-f009:**
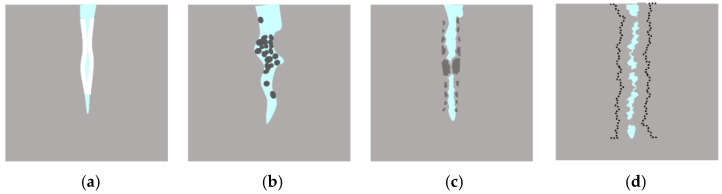
Crack-healing mechanism. (**a**) CaCO_3_ crystal. (**b**) Impurities in water or loose concrete particles. (**c**) Further hydration of the cementitious material. (**d**) C-S-H expands by absorbing water.

**Figure 10 materials-15-04518-f010:**
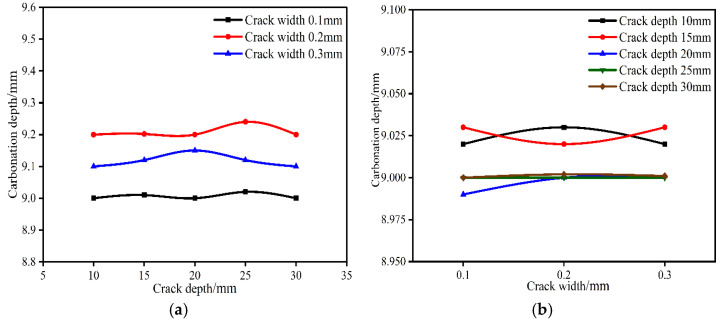
Effect of crack size on carbonation depth at the opposite side of crack face when the water–binder ratio is 0.35 and the carbonation time is 14 d. (**a**) Effect of crack depth on carbonation depth at the opposite side of the crack. (**b**) Effect of crack width on carbonation depth at the opposite side of the crack.

**Figure 11 materials-15-04518-f011:**
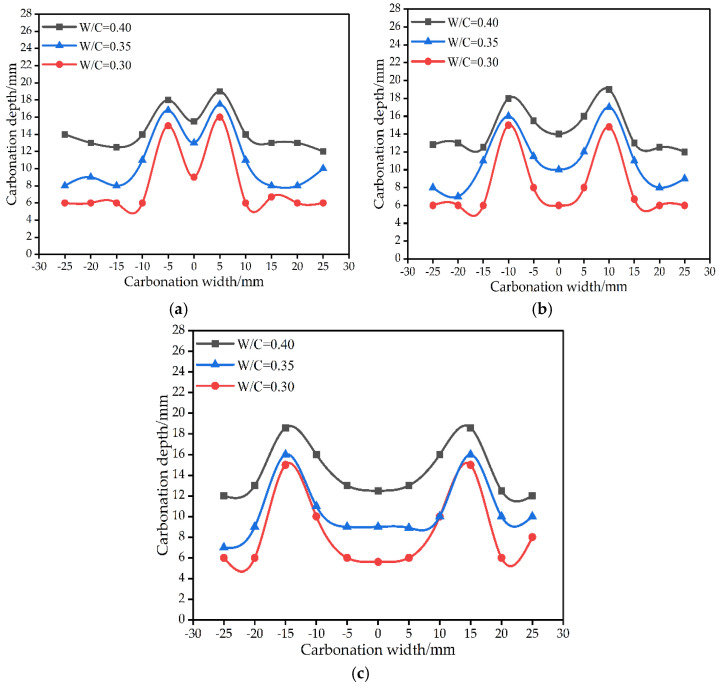
The effect of crack distance on carbonation depth when the carbonation time is 14 days. (**a**) The crack distance is 10 mm. (**b**) The crack distance is 20 mm. (**c**) The crack distance is 30 mm.

**Figure 12 materials-15-04518-f012:**
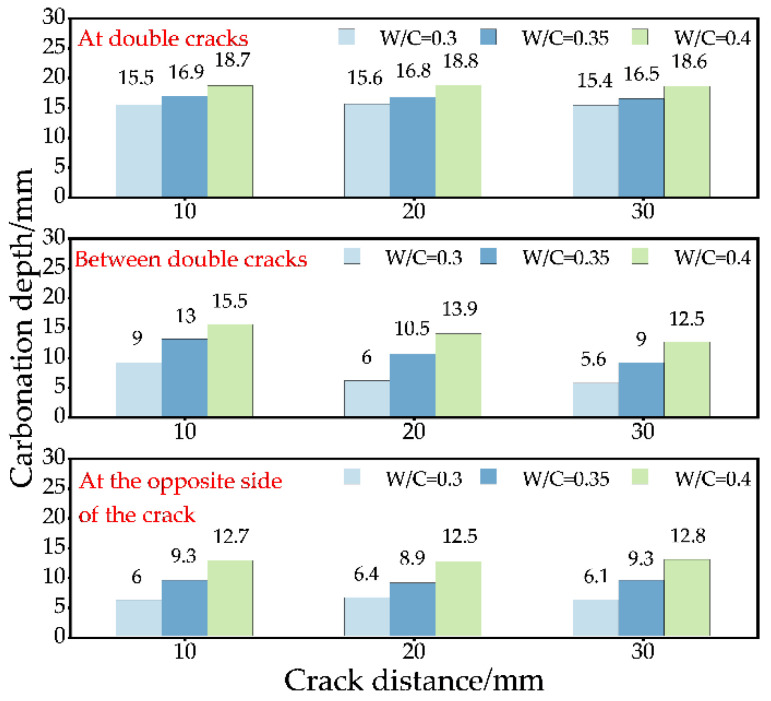
Effect of double-crack distance on carbonation process when the carbonation time is 14 days.

**Figure 13 materials-15-04518-f013:**
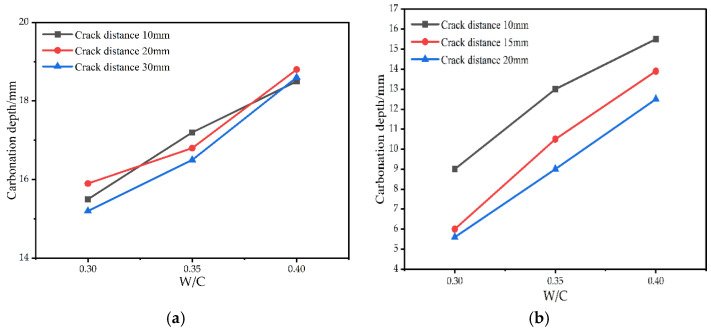
Effect of water–binder ratio on carbonation depth when the carbonation time is 14 days. (**a**) Effect of water–binder ratio on carbonation depth of the crack. (**b**) Effect of water–binder ratio on carbonation depth between cracks. (**c**) Effect of water–binder ratio on carbonation depth of the opposite side of the crack.

**Figure 14 materials-15-04518-f014:**
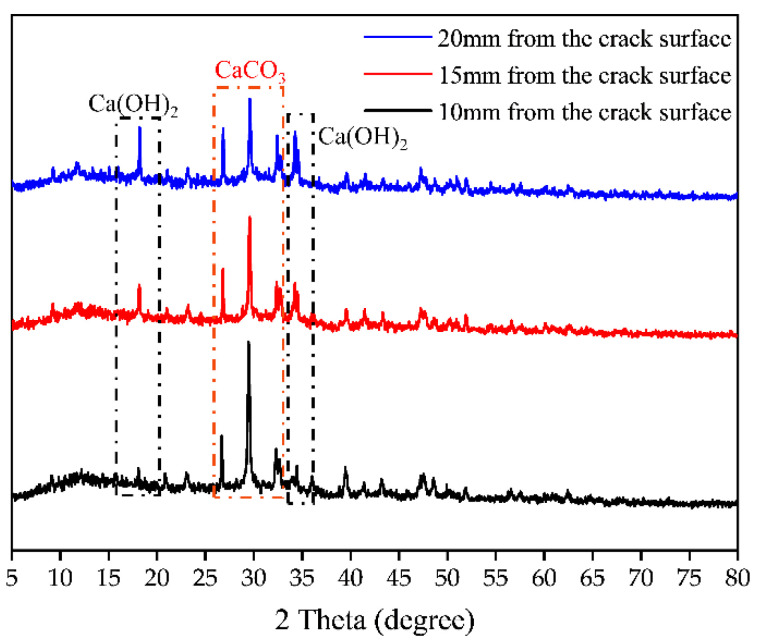
XRD phase analysis.

**Table 1 materials-15-04518-t001:** The chemical composition of P • O 42.5 normal Portland cement wt.%.

SiO_2_	Fe_2_O_3_	Al_2_O_3_	MgO	CaO	SO_3_	Na_2_O	K_2_O	Other
20.89	5.44	3.96	1.71	62.24	2.65	0.27	0.21	2.63

**Table 2 materials-15-04518-t002:** Magnification factor of carbonation depth of specimens with different carbonation time and water–binder ratios.

Carbonation Time	W/C	Carbonation Depth at Crack/mm	Carbonation Depth at Opposite Side of the Crack/mm	Amplification Factor
7d	0.3	15.5	4	3.87
7d	0.35	16	5.4	2.91
7d	0.4	16.3	8.5	1.92
14d	0.3	16.2	5.6	2.94
14d	0.35	16.8	8.5	1.97
14d	0.4	18	13.1	1.38
21d	0.3	16.6	7.3	2.27
21d	0.35	18	10.7	1.68
21d	0.4	24.9	17.5	1.44

**Table 3 materials-15-04518-t003:** Fitting table of proportion coefficient A at crack.

W/C	A	R^2^	Standard Error	t	Sig.
0.3	6.703	0.993	0.561	11.946	0.007
0.35	7.07	0.995	0.497	14.213	0.005
0.4	8.388	0.997	0.486	17.259	0.003

**Table 4 materials-15-04518-t004:** Fitting table of proportion coefficient A at the opposite side of the crack.

W/C	A	R^2^	Standard Error	t	Sig.
0.3	2.417	0.996	0.157	15.429	0.004
0.35	3.533	0.994	0.284	12.443	0.006
0.4	5.627	0.992	0.520	10.819	0.008

**Table 5 materials-15-04518-t005:** Fitting table of proportion coefficient D and the constant E.

Position	D	E	R^2^	Standard Error	t	Sig.
At crack	16.850	-	0.990	0.549	9.823	0.009
	-	1.490	-	0.754	8.372	0.015
At the opposite of the crack	32.100	-	0.985	0.884	8.259	0.017
	--	7.376	--	0.665	8.930	0.011

**Table 6 materials-15-04518-t006:** The magnification factors for specimens with different crack depths and widths.

Crack Depth	Crack Width	Carbonation Depth at Crack/mm	Carbonation Depth at Opposite Side of the Crack/mm	Amplification Factor
10	0.1	13.2	9	1.46
10	0.2	15.6	9.2	1.69
10	0.3	16.2	9.1	1.78
15	0.1	15.1	9	1.67
15	0.2	16.8	9.2	1.82
15	0.3	18.2	9.1	2
20	0.1	15.6	8.9	1.75
20	0.2	17.4	9.1	1.91
20	0.3	18.9	9.1	2.07
25	0.1	16.1	9	1.78
25	0.2	17.6	9.2	1.91
25	0.3	19.2	9.1	2.1
30	0.1	16.2	9	1.8
30	0.2	17.5	9.2	1.9
30	0.3	19	9.1	2.08

## Data Availability

Since the experiment was completed with the support of Qingdao Agricultural University, the data used to support the results of this study are available from the responsible person and the author upon request.

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
