# Peer review of "Analysis of Carbonation Behavior of Cracked Concrete"

_materials, 2022, doi:10.3390/ma15134518_

Round 1

Reviewer 1 Report

The ABSTRACT section is well–structured, is informative, can stand alone and covers the content. It briefly summarizes the studied problem. Also, the KEYWORDS are well–defined.

The paper is structured properly and have the basic structure of a typical research paper (INTRODUCTION, MATERIAL & METHODS, RESULTS & DISCUSSIONS, CONCLUSIONS, REFERENCES, etc.). The paper is well–structured and its parts are logically interconnected. Overall, this manuscript is well–written and interesting to read.

The list of REFERENCES is long and relatively well chosen. The entire bibliography is current (the oldest being from 2003)

The Tables are representative and the Figures & Graphs have good qualities. Overall, the graphic addenda is remarkable.

The INTRODUCTION section provide the necessary background information needed to understand the paper, although is quite succinctly presented. The aims and objectives of the research are well defined. The review of the literature presented in this section is, in general, adequate.

The MATERIALS & METHODS section (TEST METHODS & PRINCIPLES) is relatively well described and include detailed information. This section include all the technical details of the experimental setup, measurement procedure, materials and preparation of specimens, details on the XRD test and the rapid carbonation test of the specimens. Overall, this section is technically and fairly detailed.

The RESULTS & DISCUSSIONS section is well–structured. This section describe the important RESULTS of the research on the carbonation process of the single crack specimen and of the specimen with double crack, followed by several DISCUSSIONS upon the on–site measurements.

The CONCLUSION section succinctly summarize the major points of the paper, derived from the RESULTS & DISCUSSIONS. The authors fairly concludes in just a few sentences given the rich discussion in the body of the paper.

Author Response

Thank you very much for your comments. Those comments are all valuable and very helpful for revising and improving our paper. We have made relevant amendments based on the suggestions and comments, and we hope the paper is more readable. We uploaded files and responded to your comments point-to-point

Reviewer 2 Report

FOR AUTHORS: 

The submitted paper is interesting to show an analysis of carbonation behavior of cracked concrete under laboratory conditions. The paper is original and suitable for Materials after revision. I recommend major revision.

Comments and suggestions for authors to improve the original are as follows:

1. Introduction. This part of the submitted paper is condensed and included 21 references. The main points reviewed and considered by the authors are carbonation and cracks in concrete. This section is well structured and presented. Studies on the carbonation law of concrete at the crack, with factors affecting the behavior, are mentioned. One author must be included as "Bogas et al. [21]" instead of "Bogas".

I think that this part of the paper can be improved taking into account some precedent papers not cited by the authors, in particular the references of Van Balen and Van Gemert on modelling (1994), Campos Silva et al. on carbonation (2016) and Romero-Hermida et al. on carbonation (2021). These references are mentioned in this comments and suggestions for authors (see afterwards). 

The goal/aim of the research is clearly exposed and the method for test seems well designed. The final consideration (effective theoretical foundation for the future study of predicting the life of concrete structures) is of particular interest taking into account the results of this investigation and its potential impact.

2. Materials and Methods: I consider that this section of the submitted paper must be improved. Statistical results and additional details are necessary for a better presentation. Somme comments and suggestions are indicated.

2.1. Materials and preparation of specimens. The number of samples (specimens) prepared for the experiments must be indicated. Why copper (with its surface lubricated with oil) is used for the obtention of cracks? Is it possible the use of a distinct metal?

Table 1 shows the chemical composition. The units must be "wt. %". Please, indicate them in the Table. The total sum of the chemical composition (as oxides) is 97.37 wt. %. So the difference could be explained considering the "loss on ignition"(loss of weight after heating at 1000 ºC 1h) or the presence of additional elements. Please, explain it.

2.2 Carbonation test. The concentration of phenolphtalein solution (and the solvent, possibly ethanol and deionized water) must be indicated.

The authors write "The specimens were carbonized" and "the non carbonized area was pink". Please, revise. Is it "carbonated" and "non carbonated"? The "measured data" must be studied using statistical procedures, providing number of determinations, average of results, standard deviations, etc. It is of particular interest concerning the results of Figures 4-8 and 10-12 and additional results included in Tables 2 and 5. 

2.3" XRD test" must be "X-ray powder diffraction (XRD) test". The samples can be studied using oriented or random preparations. please, give details. More details concerning conditions of X-ray runs must be provided.

3. Results and discussion. 

The authors use "carbonation process" and "carbonization". One is concerned on the effect of CO2 on a material, in particular the reaction of CO2  to give carbonates, bicarbonates and carbonic acid. "Carbonization" suggests a thermal effect/burning/pyrolysis effect on organic substances yielding carbon as the last result. In the caption of Figure 4 the word is "carbonation" and all the indications of Y-axis and X-axis, and the indication inside the Figures, are described as "Carbonization". in Tables 2 and 5 the word is "Carbonation". Please, correct when the word be wrong for a better presentation and along all the text.

The authors (in page 5) indicate that the porosity increases and the compactness decreases. Is there additional data on these indications?

In subsection 3.1.3, I think that the reference "Zhao et al. [24]" is not correct. It must be "Shun Bo et al. [24]". Revise.

Please, explain better the symbols meaning D, C, E and W/C (water/binder ratio). I think that the equations (1) and (2) must be corrected for a better presentation using the water-binder ration in the form (W/C) to avoid any confusion with the rest of the symbols. I suggest the support of these results afteer fitting using statistical methods providing average results, standard deviations, etc.

Th depth of carbonation with time was studied using the solution of Fick's diffusion law in the form x = h (t1/2) where x is the depth (in mm), t is the time (years) and h is the advance speed of the carbonation fornt. Once h has been calculated as a function of time, depths at different ages of the samples can be predicted. See Van Balen, K; Van Gemert, D. Modelling lime mortar carbonation, Mater. Struct. 1994, 27, 393-398; Campos Silva et al., Study of carbonation behavior...Concr. Cem. Investig. Desarro. 2016, 8, 14-34. The investigations by Romero-Hermida, M.I. et al.is of particular interest, with title "Characterization and analysis of the Carbonation Process of a Lime Mortar Obtained...", Int. J. Environ. Res. Public Health 2021, 18, 6664.

The scale coefficient fitting table (Table 3) must be presented including the statistical analysis.

Discussion of results in page 7 is very interesting taking into account the chemical reactions. The deduction of the authors that "the influence of crack width on the carbonation process at the cracks is more important than the crack depth" is of particular interest.

Please, revise the label of Y-axis of Figures 7 and 8: "Carbonization" is not correct, as pointed out above These labels in Figure 10 (a, b and c) are correct. However, when the subsection 3.2.1 is mentioned, the authors indicate "Carbonization". Please, correct it. This subsection is very attractive concerning the study of double crack specimens. 

The label of Y-axis in Figure 12 b (Carbantion) must be revised and corrected. I think that additional references are important to mention in  subsection 3.3 when the reaction of CO2 with Ca(OH)2 is discussed.

The carbonation mechanism (subsection 3.3) is discussed taking into acccount the XRD results presneted in Figure 13. The authors indicate that phenolphtalein indicator is colored and moe. I think that a complementary Figure (Photographies of the sections) would be interesting showing experimental results as pictures on the effect of the bidirectional intrusion pattern along with the crack point.

References must be revised and corrected for a better presentation. All these references contains the symbol "[J]" after the titles are finished. Reference 22 contains "[S]" after the title. 

Reference 20 must be corrected in a word of the title "Carbonation" instead of "Crabonation".

Revise reference 27: the year is not 2011. It must be 2012.

Taking into account the comments and suggestions, the authors can prepare a new version of this original paper for further revision.

Author Response

Thank you very much for your comments. Those comments are all valuable and very helpful for revising and improving our paper. We have made relevant amendments based on the suggestions and comments, and we hope the paper is more readable.We uploaded files and responded to your comments point-to-point.

Reviewer 3 Report

The presented paper dealing with the carbonation of cement pastes does not bring any new findings to the monitored issues. Cement carbonation is already studied in great detail, including all sorts of influences, carbonation rate models depending on various conditions and damage to the cement matrix, etc. For this reason, I recommend rejecting the paper for publication. The paper is not very professional to be published in an impact journal. 

Further comments are below:

It is absolutely necessary for the next time to replace the word "carbonized" with "carbonated" and "carbonization" for "carbonation" throughout the paper. 

The carbonation time (age of the samples) should be indicated for all Figures.

Figures 7 and 8 have swapped parts (a) and (b).

The authors report that they performed carbonation tests using phenolphthalein solution. It would be useful to supplement the paper with some images from these tests for better illustration.

L 293: The statement is incorrect - phenophthalein is not colored in carbonated cement, but vice versa.

Author Response

Thank you very much for your comments. Those comments are all valuable and very helpful for revising and improving our paper. We have made relevant amendments based on the suggestions and comments, and we hope the paper is more readable. We uploaded files and responded to your comments point-to-point.

Round 2

Reviewer 2 Report

The authors revised the submitted original manuscript properly according to my previous comments and suggestions. Statistical treatment of data has been perfomed.

Additional experimental details have been included. Equations in page 7 have been revised and amended.

According to all these changes, I consider that the paper can be accepted. However, I recommend minor revision because some corrections must be performed before final acceptance.

The comments and suggestions are as follows:

Table 1 must be corrected for a better presentation. Please, indicate the meaning of "R" in "R2O (R=Na, K ???). The authors include "SiO" at the first column. It must be "SiO2". Please, correct it.

In Table 2, the heading includes two times "carbonization".It must be "carbonation". Please, correct because it is a mistake. All the text must be revised to avoid the use of "carbonization" and "carbonized". The paper focus on the carbonation behavior of cracked concrete. For instance, revise page 13, final paragraph. The word "carbonized" must be changed to "carbonated" (from carbonation). 

Additional references included in the revised version must be checked (references 11 and 17). 

Reference 11. Authors must be written as follows: K. Van Balen, D. Van Gemert.

Reference 15: the journal must be abreviated as Concr. Cem. Investig. Desarro. 

Reference  17: the journal must be abreviated as Int. J. Environ. Res. Public Health.

The authors in this reference 17 must be written as follows:

M.I. Romero-Hermida, A.M. Borrero-López, V. Flores-Alés, F.J. Alejandre, J.M. Franco, A. Santos, L. Esquivias.

Then, it can be checked in the scientific literature that these authors signed the papers using this form (cited as Romero-Hermida et al.).

A final English review of the text is recommended.

Author Response

Thanks for giving us the chance to revise our manuscript. The manuscript has been carefully revised according to your comments and the itemized response is attached. The second changes made were highlighted in pink font. We now submit our revised manuscript for your kind re-consideration.

Reviewer 3 Report

Despite some improvement in the quality of the paper after the reviewers' comments, the paper remains of poor quality, mainly in terms of content. The opponent does not want to oppose the publication of this paper at all costs, in his opinion with facts known for a long time, and therefore leaves the decision to the editor and other reviewers. Only two short comments about the authors:

Please change the terms "carbonized" (line 51, 82, 314) and "carbonization" (label of Table 2) to "carbonated and carbonation." The meaning of the word carbonized is completely different from carbonated.

Table 1, it would be better to explain the meaning of R2O in a note.

Author Response

(The authors gave the same response as above.)
